# A comparative framework for convergence analysis of perturbation series techniques in nonlinear fractional quadratic differential equations

**Dulfikar Jawad Hashim** [ID]*

Mathematics Department, Faculty of Computer Science and Mathematics, University of Thi-Qar, Thi-Qar, Iraq

* dulfikar_math91@utq.edu.iq

## Abstract

This study tackles the challenge of obtaining highly accurate approximate solutions for nonlinear fractional differential equations, which often lack exact solutions due to their inherent complexity. A unified perturbation framework is proposed based on homotopy topology theory, enabling multiple formulations depending on the number of convergence-control parameters. Through dynamic adjustment of these parameters, the Homotopy Method achieves enhanced precision, particularly for fractional-order models exhibiting long-memory behavior. Numerical results clearly demonstrate that increasing the number of convergence parameters leads to significantly improved accuracy. Supported by detailed graphs and tables, the proposed approach proves to be a flexible, robust, and reliable tool for solving nonlinear fractional differential equations.

## Introduction

Recently applied mathematics plays a pivotal tool for modeling several models of real world application, especially in computer science [1–4], engineering [5], artificial intelligence [6], communication [7–8], and deep learning [9–10]. Fractional Differential Equations (FDEs) have recently been employed to describe the dynamics of several nonlocal complex systems based on the long-memory property, for instance, in fluid mechanics, chemistry, viscoelasticity, control theory, and several physical and engineering problems. Such systems cannot be accurately modeled by classical differential equations, especially for real-world phenomena that inherently involve memory effects [11].

In contrast to classical differential equations, FDEs are formulated using several fractional derivative definitions such as the Riemann–Liouville and Caputo derivatives. The Caputo fractional derivative is considered a more advanced form of the Riemann–Liouville derivative, making it more convenient for handling real

**Data availability statement:** All relevant data are within the paper.

**Funding:** The author(s) received no specific funding for this work.

**Competing interests:** The authors have declared that no competing interests exist.

applications, particularly in dynamic systems that are difficult to model using the Riemann–Liouville formulation [12].

The Perturbation Series Method (PSM) is regarded as a powerful technique, constructed independently of small or large parameters, to approximate analytical solutions of nonlinear problems. However, only HAM [13] and OHAM [14] provide a systematic procedure to control the convergence region of the series solution through the introduction of a convergence-control parameter. In other words, when solving strongly nonlinear problems that may lead to divergent solutions, overcoming such obstacles is one of the major challenges facing numerical techniques; in this case, HAM or OHAM can be employed to adjust the convergence of the series solution.

Despite the advantages of HAM and OHAM in adjusting the convergence of series solutions, these methods still suffer from certain limitations. In particular, their reliance on a single convergence-control parameter often restricts flexibility and may lead to suboptimal accuracy when dealing with highly nonlinear fractional systems. This highlights the need for a more general perturbation framework that incorporates multiple convergence parameters, thereby offering improved adaptability and precision. Addressing this gap is of significant importance, as accurate solutions of fractional differential equations play a crucial role in modeling real-world phenomena with memory effects in science and engineering.

The Italian nobleman Count Jacopo Francesco Riccati (1676–1754) introduced one of the most important quadratic nonlinear differential equations, which has been widely used to model several real applications in mathematical finance, engineering, and applied sciences, such as damping laws, rheology, diffusion processes, transmission line phenomena, and optimal control theory problems [15].

In this work, we introduce a new dynamic framework based on the Perturbation Series Method (PSM) for investigating the convergence behavior of the numerical solution of the quadratic fractional differential equation (QFDE) of the following form:

$$\frac{d^{\mathcal{B}}}{dt^{\mathcal{B}}}\, x(t) + a_1(t)x^2(t) + a_2(t)x(t) + a_3(t) = 0, \quad \mathcal{B} \in (0,1],\ t \in [t_0, T] \tag{1}$$

Subject to the following supplementary initial condition.

$$x(t_0) = 0 \tag{2}$$

Here, $a_1(t)$, $a_2(t)$, and $a_3(t)$ are real continuous functions, and $\frac{d^{\mathcal{B}}}{dt^{\mathcal{B}}}$ represent the nonlocal fractional Caputo derivative. to present the main definitions and theories that help to handle this work see [6].

The outline of this research paper will be arranged as starting the introduce the general fractional perturbation structure in Section two, while Section three will present the application of fractional perturbation techniques for solving QFDE, finally we will present the conclusions and recommendation in section four.

## 2. PSM Technniqe

In this section, we will introduce a new PSM form involve HPM [7], HAM, and OHAM starting from employs the notion of the homotopy from topology to construct a convergent series solution as follows:

Consider QFDE in Eq. (1), followed by the nonlinear form of order $\mathcal{B} \in [n-1, n]$, for n $\in$ N

$$f[x(t)] = \frac{d^{\mathcal{B}}}{dt^{\mathcal{B}}} x(t) + a_1(t)x^2(t) + a_2(t)x(t) + a_3(t),$$

(3)

here, $f$ is a nonlinear fractional differential operator, $t$ is an independent variable, and $x(t)$ is an unknown function of the independent variable $t$, and $\frac{d^{\mathcal{B}}}{dt^{\mathcal{B}}} = D_0^{\mathcal{B}}$ described the Capotu fractional derivative in [16], then based on the approximate homotopy perturbation techniques were developed based on the notion of homotopy in topology between two continuous functions based on the deformation parameter $p$ as follows:

$$\varnothing(t; p) : [t_0, T] \times p \rightarrow \mathbb{R}$$

(4)

According to [17,18], and [19], we can construct the general overall perturbation zeroth order deformation equation as below:

$$\begin{cases} HPM(t; p) = (1-p) \ \mathcal{L} \left[\varnothing(t; p)_{HPM} - x_0(t)\right] - HPM(p)f\left[\varnothing(t; p)_{HPM}\right], \\ HAM(t; p) = (1-p) \ \mathcal{L} \left[\varnothing(t; p)_{HAM} - x_0(t)\right] - HAM(p)f\left[\varnothing(t; p)_{HAM}\right], \\ OHAM(t; p) = (1-p) \left[\mathcal{L} \left(\varnothing(t; p)_{OHAM}\right)\right] - OHAM(p)f\left[\varnothing(t; p)_{OHAM}\right], \end{cases}$$

(5)

here, $\mathcal{L} = D^{\beta}$ indicate to Caputo fractional derivative, $\varnothing(t; p)_{HPM}$, $\varnothing(t; p)_{HAM}$, and $\varnothing(t; p)_{OHAM}$ are the unknown functions that must be satisfied in the initial condition, $HPM(P), HAM(p)$, and $OHAM(p)$ refer to the auxiliary convergence functions, $x_0(t)$ is the initial guess of the approximate solution $x(t)$, and $0 \le p \le 1$ is the embedding parameter, such that embedding parameter generate the approximate perturbation series solution when it is deforming from zero to one.

According to Eq. (5), we can conclude that OHAM series built independent of the initial approximation while for HPM and HAM one can construct the initial approximation series by setting $p = 0$ in Eq. (5) as follows:

$$\begin{cases} HPM(t; 0) = \ \mathcal{L} \left[\varnothing(t; 0)_{HPM} - x_0(t)\right], \\ HAM(t; 0) = \ \mathcal{L} \left[\varnothing(t; 0)_{HAM} - x_0(t)\right]. \end{cases}$$

(6)

On the other hand, the exact solution via HPM, HAM, and OHAM will be formulated by setting $p = 1$, as follows:

$$\begin{cases} HPM(t; 1) = f\left[\varnothing(t; 1)_{HPM}\right], \\ HAM(t; 1) = -hf\left[\varnothing(t; 1)_{HAM}\right], \\ OHAM(t; 1) = -\sum_{j=1}^{\infty} c_j \left[f\left[\varnothing(t; 1)_{OHAM}\right]\right]. \end{cases}$$

(7)

Eq. (7) employ to find the exact analytical solution of fractional differential equation via PSM for $p = 1$, in other word that is mean we must formulate infinite PSM series and that will be impossible especially for nonlinear cases.

When $p$ varying from zero to one, the solution $\varnothing(t; p)$ changing from the initial guess $x_0(t)$ to the exact solution $x(t, 1)$. Now, we will expand the approximate solution of PSM $\varnothing(t; p)$ as a Tylor series about $p$ to obtain the following PSM series solution:

$$\begin{cases} \varnothing(t;p)_{HPM} = x_0(t) + \sum_{j=1}^k x_j(t)p^j, \\ \varnothing(t;p)_{HAM} = x_0(t) + \sum_{j=1}^k x_j(t)p^j, \\ \varnothing(t;p)_{OHAM} = x_0(t) + \sum_{j=1}^k x_j(t;c_j)p^j. \end{cases} \tag{8}$$

Such that, for HPM and HAM we will construct the first until $k^{th}$ order of series as follows:

From Eq. (8) we have

$$x_j(t) = \frac{1}{j!} \left. \frac{\partial^j \widetilde{\varnothing}(t;p)}{\partial p^j} \right|_{p=0}. \tag{9}$$

Now by defining the vectors in Eq. (10), it is possible to deduce the governing equations from the zero-order deformation in Eq. (10) as in

$$\overrightarrow{x_k}(t) = \{x_0(t), x_1(t), \dots, x_j(t)\}. \tag{10}$$

For $k$ time derivatives of Eq. (10) in terms of $q$, and then set $q = 0$, followed by dividing them by $k!$, we obtain the $k^{th}$-order deformation equation

$$\mathcal{L}[x_k(t) - \psi_k x_{k-1}(t)] = h\mathcal{R}_k\left(\overrightarrow{x_{k-1}}(t)\right) \tag{11}$$

Followed by employ Riemann-Liouville integral concept in [6] and utilize the property of Caputo derivative we can reformulate Eq. (11) in the following form

$$x_k(t) = \psi_k x_{k-1}(t) + \mathcal{J}_0^B h\mathcal{R}_k\left(\overrightarrow{x_{k-1}}(t)\right) \tag{12}$$

where

$$\mathcal{R}_m\left(\overrightarrow{y_{k-1}}(x)\right) = \frac{1}{(k-1)!} \left. \frac{\partial^{k-1}\mathcal{N}_f[\vartheta(x;q)]}{\partial q^{k-1}} \right|_{p=0}, \psi_k = \begin{cases} 0, k \leq 1 \\ 1, k > 0 \end{cases}. \tag{13}$$

Here, Eq. (12) employ to approximate the analytical solution of nonlinear differential equation via PSM technique.

PSM method have different accuracy level depending on the auxiliary convergence functions mentioned Eq. (5) such that we can define the convergence functions of HPM, HAM, and OHAM as below in Eq. (14)

$$\begin{cases} HPM(p) = -p, \\ HAM(p) = ph, \\ OHAM(p) = \sum_{j=1}^k c_j p^j = c_1 p + c_2 p^2 + \dots + c_k p^k. \end{cases} \tag{14}$$

Such that, for HPM is a special case of HAM for $h = -1$, while HAM have different values of $h$, and for OHAM provide many convergence control parameters at each series term and that will help us to control the convergence area of PSM series based on the residual form of Eq. (8).

In this research paper we will find the convergence parameters at each different value of the pair $\{t, \mathcal{B}\}$, that means we must find the convergence parameters at each different value of $t$ for different value of fractional order B.

Now, by employing Eq. (7), Eq. (8), and Eq. (14) into Eq. (5) we can formulate the $k^{th}$ order PSM approximate series solution as follows:

$$\begin{cases} HPM = x_0(t) + \sum_{j=1}^{k} x_j(t), \\ HAM = x_0(t) + \sum_{j=1}^{k} x_j(t), \\ OHAM = x_0(t) + \sum_{j=1}^{k} x_j(t; c_j). \end{cases} \tag{15}$$

Here, Eq. (15) will be use in order to find the approximate solutions of the nonlinear fractional differential equation under the Caputo fractional derivative.

Now we will move to the practical section to show the capability of the above algorithm for solving QFDE to refute the idea that say HPM, and HAM provide same accuracy for QFDE.

## 1. Numerical experiments

To give a clear overview of PSM technique, we present the following illustrative Applications, including first and second order QFDE

**Application 1:**

Consider the following first order QFDE [19].

$$\begin{cases} D_0^{\mathcal{B}} x(t) + x^2(t) - 1 = 0, & \mathcal{B} \in (0, 1] \\ x(0) = 0. \end{cases} \tag{16}$$

Subject to the following exact solution, when $\mathcal{B} = 1$

$$exact = \frac{e^{2t} - 1}{e^{2t} + 1} \tag{17}$$

Firstly, and according to Section 2 and based on Eq. (5), we can construct the sixth-order PSM series solution of Eq. (16) as follows:

$$\begin{cases} HPM(t; p) = (1 - p) \, D_0^{\mathcal{B}} \left[ x_0(t) + \sum_{j=1}^{6} x_j(t)p^j - 0 \right] + pf \left[ x_0(t) + \sum_{j=1}^{6} x_j(t)p^j \right], \\ HAM(t; p) = (1 - p) \, D_0^{\mathcal{B}} \left[ x_0(t) + \sum_{j=1}^{6} x_j(t)p^j - 0 \right] - phf \left[ x_0(t) + \sum_{j=1}^{6} x_j(t)p^j \right], \\ OHAM(t; p) = (1 - p) D_0^{\mathcal{B}} \left[ x_0(t) + \sum_{j=1}^{6} x_j(t; c_j)p^j \right] - \sum_{j=1}^{k} c_j p^j \, f \left[ x_0(t) + \sum_{j=1}^{6} x_j(t; c_j)p^j \right], \end{cases} \tag{18}$$

Now, by applying Riemann-Liouville integration and based on the property of Caputo derivative we can rewrite the PSM form as below

$$\begin{cases} HPM(t; p) = (1 - p) \left[ \sum_{j=1}^{6} x_j(t)p^j - 0 \right] + p\mathcal{J}_0^{\mathcal{B}} f \left[ \sum_{j=1}^{6} x_j(t)p^j \right], \\ HAM(t; p) = (1 - p) \left[ \sum_{j=1}^{6} x_j(t)p^j - 0 \right] - ph\mathcal{J}_0^{\mathcal{B}} f \left[ \sum_{j=1}^{6} x_j(t)p^j \right], \\ OHAM(t; p) = (1 - p) \left[ \sum_{j=1}^{6} x_j(t; c_j)p^j \right] - \sum_{j=1}^{k} c_j p^j \mathcal{J}_0^{\mathcal{B}} f \left[ \sum_{j=1}^{6} x_j(t; c_j)p^j \right], \end{cases} \tag{19}$$

Secondly, we will utilize the algorithm in Section 2, and based on the residual error of Eq. (16), we will find the optimal convergence parameters of PSM series as follows:

For HAM case we will plot the valid region of the optimal convergence parameter $h$ at different value of $t \in [0.1, 0.4]$

This figure illustrates the range of admissible values of the convergence-control parameter $h$, showing where the homotopy analysis method yields a stable and convergent series solution of Eq. (16).

According to Fig 1, the valid region of $h$ bounded by $h \in [-1.2, -0.6]$ then we collect the optimal value of the convergence control parameter such that $h = -0.923293499986332$. On the other hand, OHAM provide more strong procedure to control the convergence area such that we have (Table 4),

In the next step, we shall employing the convergence control parameter that extracted based on the minimum residual of HAM series which is $h = -0.923293499986332$, and the convergence control parameters listed in Table 1 for OHAM series in order to optimize the solution of QFDE of order $\mathcal{B} = 1$, on the other hand, for solving QFDE of order $\mathcal{B} = 0.8$, we will use $h = -0.923293499986332$, and the convergence control parameters listed in Table 2 for HAM series and OHAM series respectively (Fig 2).

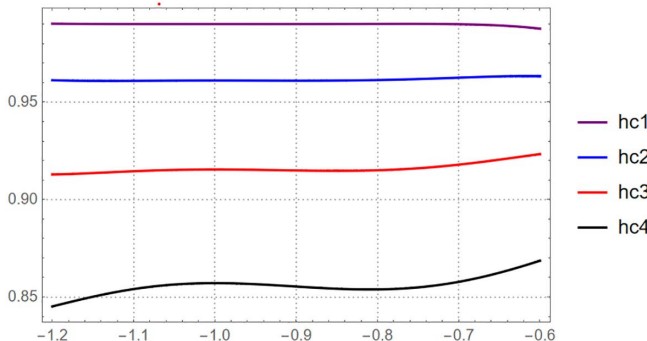

**Fig 1. The valid region of $h$-curve for solving Eq. (16) via sixth-order HAM series.**

**Table 1. The optimal convergence parameter of sixth-order OHAM for solving Eq. (16) of order $\mathcal{B} = 1$ for different values of $t$.**

| $c_j$ $\diagdown$ $t$ | 0.1 | 0.2 | 0.3 | 0.4 |
|---|---|---|---|---|
| $c_1$ | −1 | −1 | −1 | −1 |
| $c_2$ | 0.00076778885 | 0.00657302979 | 0.01434900752 | 0.02448128757 |
| $c_3$ | −0.00002555801 | 0.00053594436 | 0.00115274736 | 0.00114759537 |
| $c_4$ | −0.00000375101 | 0.00002159674 | 0.00011149053 | 0.00032111415 |
| $c_5$ | 0.00071436857 | −0.00068697331 | −0.00458526515 | −0.01731270484 |
| $c_6$ | $-7.72881801 \times 10^{-9}$ | $-3.66554388 \times 10^{-7}$ | −0.00000215622 | −0.00000351585 |

**Table 2. The optimal convergence parameter of sixth-order OHAM for solving Eq. (16) of order $\mathcal{B} = 0.8$ for different values of $t$.**

| $c_j$ | $t$ | | | |
|---|---|---|---|---|
| | 0.1 | 0.2 | 0.3 | 0.4 |
| $c_1$ | −1 | −1 | −1 | −1 |
| $c_2$ | 0.00697623486 | 0.02005647783 | 0.03582368188 | 0.05244507261 |
| $c_3$ | −0.00028844400 | −0.00223209582 | −0.00681686403 | −0.01704456012 |
| $c_4$ | 0.00000583853 | 0.00002187698 | −0.00000493255 | −0.00042230190 |
| $c_5$ | −0.00004451091 | −0.00079039971 | −0.00727930420 | −0.01774076728 |
| $c_6$ | $-7.70645312265 \times 10^{-8}$ | −0.00000288135 | −0.00002185049 | −0.00014046175 |

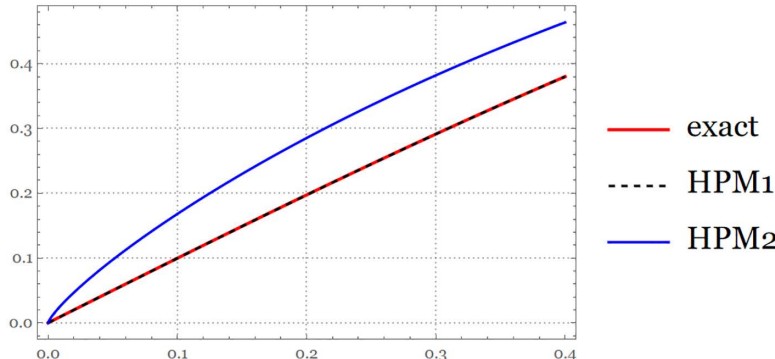

**Fig 2. The exact solution and approximate solution via HPM series at $\mathcal{B} = 1$, and $\mathcal{B} = 0.8$ for $t \in [0, 0.4]$.**

$$\begin{cases} \text{exact} = \frac{e^{2t}-1}{e^{2t}+1}, \\ \text{HPM 1} = t - \frac{1}{3}\,t^3 + \frac{2}{15}\,t^5, \\ \text{HPM 2} = 1.0736712740308256\,t^{0.8} - 0.5528058354449037\,t^{2.4} + 0.3836534567099019\,t^4. \end{cases} \tag{20}$$

The figure above compares the HPM approximate solutions with the exact solution, illustrating the accuracy and convergence of the method over the considered time interval (Fig 3).

$$\begin{cases} \text{exact} = \frac{e^{2t}-1}{e^{2t}+1}, \\ \text{HAM 1} = t - 0.33318079840135617\,t^3 + 0.12377330643712081\,t^5, \\ \text{HAM 2} = 1.0736710553230022\,t^{0.8} - 0.5525528688433852\,t^{2.4} + 0.35614542647261516\,t^4. \end{cases} \tag{21}$$

The figure above demonstrates the accuracy of the HAM approximate solutions by comparing them with the exact solution over the specified time interval (Fig 4).

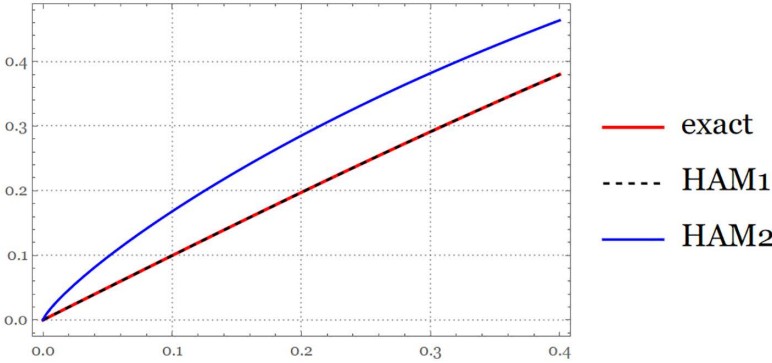

**Fig 3. The exact solution and approximate solution via HAM series at $\mathcal{B} = 1$, and $\mathcal{B} = 0.8$ for $t \in [0, 0.4]$.**

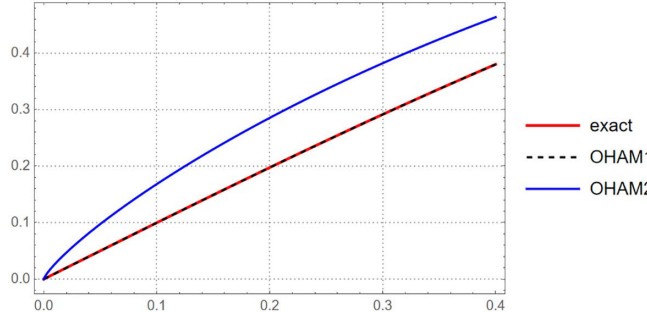

**Fig 4. The exact solution and approximate solution via OHAM series at $\mathcal{B} = 1$, and $\mathcal{B} = 0.8$ for $t \in [0, 0.4]$.**

$$\text{exact} = \frac{e^{2t} - 1}{e^{2t} + 1},$$

$$\begin{cases} OHAM\ 1 = t - 0.33333586603308873\ t^3 + 0.13282147410125267\ t^5, \\ OHAM\ 2 = 1.0736708154490522\ t^{0.8} - 0.5526278674942091\ t^{2.4} + 0.3702711736147932\ t^4, \end{cases} \quad t \in [0, 0.1]$$

$$\begin{cases} OHAM\ 1 = t - 0.3332322780480119\ t^3 + 0.12895131347691013\ t^5, \\ OHAM\ 2 = 1.0736594138075581\ t^{0.8} - 0.5512823685582582\ t^{2.4} + 0.34517977147167844\ t^4, \end{cases} \quad t \in [0.1, 0.2]$$

$$\begin{cases} OHAM\ 1 = 0.9999946734646485\ t - 0.3328421515360098\ t^3 + 0.12376732831493775\ t^5, \\ OHAM\ 2 = 1.0735958598239357\ t^{0.8} - 0.5477219984300632\ t^{2.4} + 0.3149340597913395\ t^4, \end{cases} \quad t \in [0.2, 0.3]$$

$$\begin{cases} OHAM\ 1 = 0.9999718038436889\ t - 0.3318648507074897\ t^3 + 0.11701247495789047\ t^5, \\ OHAM\ 2 = 1.073402846134119\ t^{0.8} - 0.5413386099049333\ t^{2.4} + 0.2830497897310589\ t^4. \end{cases} \quad t \in [0.3, 0.4]$$

$$(22)$$

The figure above illustrates how the OHAM approximate solutions compare with the exact solution, highlighting the convergence and accuracy of the method over the considered time interval (Fig 5).

For more illustrate, Table 3 shows the accuracy of the given problem based on the different PSM series
The figure above compares each approximate solution with the exact solution, demonstrating the accuracy and convergence of the respective methods over the considered time interval.

**Application 2:**
consider the following second order QFDE [19].

$$\begin{cases} \frac{d^{\mathcal{B}} x}{dt^{\mathcal{B}}} + x^2(t) - 1 = 0, & \mathcal{B} \in (1, 2] \\ x(0) = 0. \end{cases}$$

$$(23)$$

For HAM case we will plot the valid region of the optimal convergence parameter $h$ at different value of $t \in [0.1, 0.4]$

The figure illustrates the range of admissible values of the convergence-control parameter $h$, showing where the sixth-order homotopy analysis method produces a stable and convergent series solutio

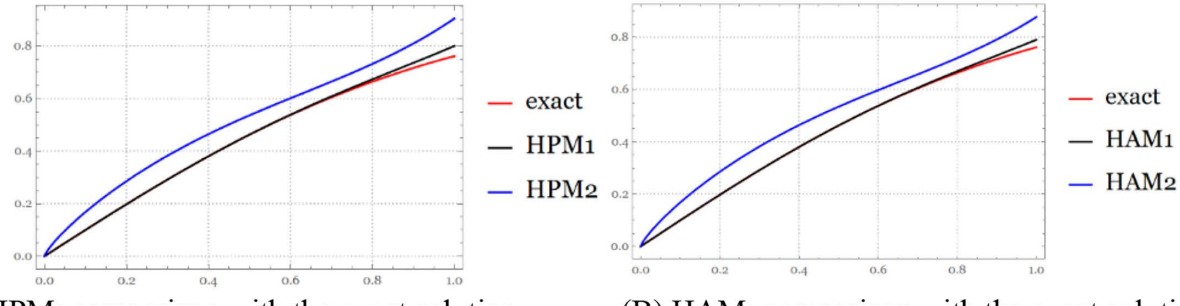

(A) HPM: comparison with the exact solution. (B) HAM: comparison with the exact solution.

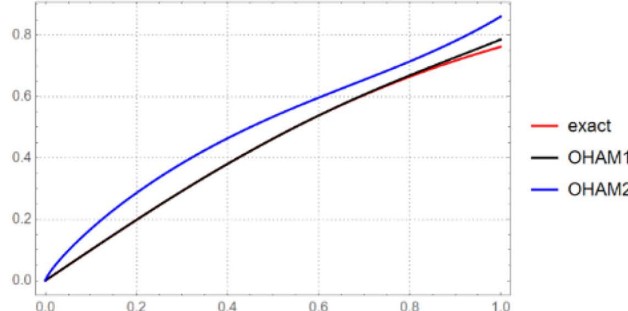

(C) OHAM: comparison with the exact solution.

**(A)**

**Fig 5. The exact solution and approximate solution via PSM series at $\mathcal{B} = 1$, and $\mathcal{B} = 0.8$ for $t \in [0, 1]$.**

**Table 3. The residual error of Eq. (21) via PSM series for different values of fractional order $\mathcal{B}$, and for $t \in [0.1, 0.4]$.**

| $\mathcal{B} = 1$ | | | |
|---|---|---|---|
| $t$ | Residual HPM | Residual HAM | Residual OHAM |
| 0.1 | $3.769 \times 10^{-7}$ | $-2.349 \times 10^{-8}$ | $5.608 \times 10^{-8}$ |
| 0.2 | $2.395 \times 10^{-5}$ | $-3.517 \times 10^{-5}$ | $2.610 \times 10^{-7}$ |
| 0.3 | $2.697 \times 10^{-4}$ | $-8.770 \times 10^{-5}$ | $2.795 \times 10^{-6}$ |
| 0.4 | $1.491 \times 10^{-3}$ | $2.738 \times 10^{-4}$ | $1.443 \times 10^{-5}$ |
| $\mathcal{B} = 0.8$ | | | |
| $t$ | Residual HPM | Residual HAM | Residual OHAM |
| 0.1 | $1.773 \times 10^{-5}$ | $-8.744 \times 10^{-6}$ | $2.663 \times 10^{-7}$ |
| 0.2 | $4.848 \times 10^{-4}$ | $2.093 \times 10^{-4}$ | $6.681 \times 10^{-6}$ |
| 0.3 | $3.310 \times 10^{-3}$ | $2.239 \times 10^{-3}$ | $4.085 \times 10^{-5}$ |
| 0.4 | $1.278 \times 10^{-2}$ | $9.947 \times 10^{-3}$ | $1.390 \times 10^{-4}$ |

According to Fig 6, the valid region of $h$ bounded by $h \in [-1.4, -0.6]$ then we collect the optimal value of the convergence control parameter such that $h = -0.9999999989$. On the other hand, OHAM provide more strong procedure to control the convergence area such that we have

For more illustrate, Table 5 and Fig 7 shows the accuracy of the given problem based on the different PSM series at $\rho = 1.9$

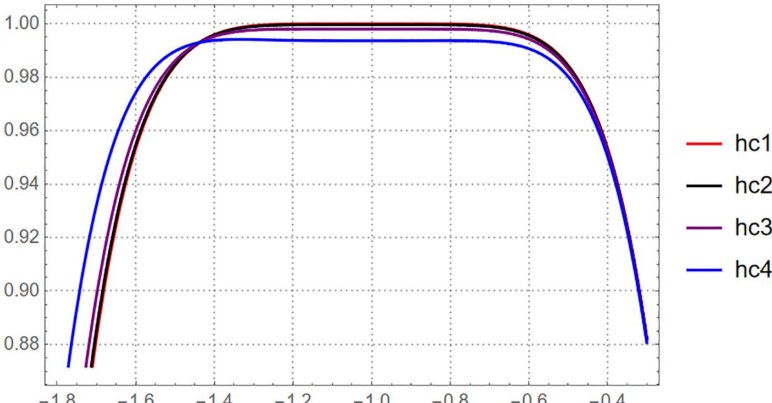

**Fig 6. The valid region of $h$-curve for solving Eq. (23) via sixth-order HAM series.**

**Table 4. The optimal convergence parameter of fifth-order OHAM for solving Eq. (23) of order $\mathcal{B} = 1.9$.**

| $c_j$ \ $t$ | 0.1 | 0.2 | 0.3 | 0.4 | 0.5 |
|---|---|---|---|---|---|
| $c_1$ | −1 | −1 | −1 | −1 | −1 |
| $c_2$ | $7.3235434 \times 10^{-7}$ | $1.1075809 \times 10^{-6}$ | $5.2000265 \times 10^{-6}$ | $1.5542549 \times 10^{-5}$ | $4.0379379 \times 10^{-5}$ |
| $c_3$ | $-1.0810565 \times 10^{-11}$ | $-2.0980686 \times 10^{-9}$ | $-4.5719550 \times 10^{-8}$ | $-4.0650511 \times 10^{-7}$ | $-2.2086355 \times 10^{-6}$ |
| $c_4$ | $-1.3232014 \times 10^{-10}$ | $-2.1409503 \times 10^{-6}$ | $-4.6293028 \times 10^{-5}$ | $1.0372028$ | $-3.3811276$ |
| $c_5$ | $-1.7958304 \times 10^{-15}$ | $-3.2883805 \times 10^{-13}$ | $-3.3277357 \times 10^{-11}$ | $-4.2119957 \times 10^{-7}$ | $1.1243729 \times 10^{-5}$ |

**Table 5. The residual error of Eq. (23) via PSM series for different values of fractional order $\mathcal{B}$, and for $t \in [0.1, 0.4]$.**

$\mathcal{B} = 1.9$

| $t$ | Residual HAM | Residual OHAM |
|---|---|---|
| 0.1 | $-1.287 \times 10^{-12}$ | $2.876 \times 10^{-15}$ |
| 0.2 | $-4.164 \times 10^{-10}$ | $5.056 \times 10^{-13}$ |
| 0.3 | $-8.955 \times 10^{-9}$ | $5.133 \times 10^{-11}$ |
| 0.4 | $-7.234 \times 10^{-8}$ | $1.361 \times 10^{-9}$ |
| 0.5 | $-3.101 \times 10^{-7}$ | $1.723 \times 10^{-8}$ |

The figure compares the HAM and OHAM approximate solutions, illustrating the accuracy and convergence behavior of both methods over the considered time interva

## 6. Conclusion

This research paper presented the PSM method based on the principle of topology to generate approximate series to solve nonlinear equations based on the long memory principle, whose exact solutions cannot be found based on traditional methods. The proposed method provides solutions in the form of series governed by the control parameters of the series convergence of solutions with different degrees of accuracy. The proposed convergence dynamics has been shown

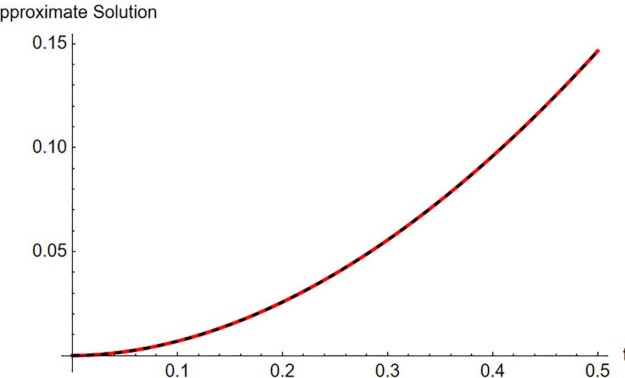

**Fig 7. The approximate via fifth-order HAM and fifth-order OHAM series at** $\mathcal{B} = 1.9$, **for** $t \in [0, 0.5]$**.**

to be effective in controlling the convergence of approximate solutions by providing approximate series with different number of convergence coefficients. The proposed method provided three types of series that differ in terms of the number of convergence parameters. The numerical results proved that increasing the convergence coefficients in the approximate series provides more accurate results as a result of the dynamics of convergence.

## Author contributions

**Methodology:** Dulfikar Jawad Hashim.

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
