## [Decision Letter · Decision Letter 0]

4 Nov 2025

Dear Dr. Hashim,

Thank you for submitting your manuscript to PLOS ONE. After careful consideration, we feel that it has merit but does not fully meet PLOS ONE’s publication criteria as it currently stands. Therefore, we invite you to submit a revised version of the manuscript that addresses the points raised during the review process.

We look forward to receiving your revised manuscript.

Kind regards,

Mohammed Jasim Mohammed Alfahdawi

Academic Editor

PLOS ONE

Journal Requirements:

2. We note that your Data Availability Statement is currently as follows: All relevant data are within the manuscript and its Supporting Information files

5. Please ensure that you refer to Figures 2, 3, and 4 in your text as, if accepted, production will need this reference to link the reader to the figures.

6. We note you have included a table to which you do not refer in the text of your manuscript. Please ensure that you refer to Table 4 in your text; if accepted, production will need this reference to link the reader to the Table.

Additional Editor Comments:

Revewer 1:

I have reviewed your paper " A Comparative Framework for Convergence Analysis of Perturbation Series Techniques in Nonlinear Fractional Quadratic Differential Equations" and I have some notes and suggestions as the following:

1-The manuscript presented in an intelligent form and written in clear and standard English and the methodology is clear

2-The abstract is clear, but it could be shortened.

3-In page 3 some symbols have been written in two way, such as p is inconsistently written as ρ and p in this paper.

4-Recommended adding more practical, examples from physical or engineering.

5-Remove the duplicates from the references, [1] and [4] are duplicates.

6-Standardizing the way of references are written must choose one format.

Finally this paper introduced a distinguished scientific contribution in the field of fractional differential equations.

,

Therefore I recommend acceptance after minor revisions.

Revewer 2:

" A Comparative Framework for Convergence Analysis of Perturbation Series Techniques in Nonlinear Fractional Quadratic Differential Equations"

The abstract is clear, but it could be shortened.

In page 3 some symbols have been written in two way, such as p is inconsistently written as ρ and p in this paper.

Recommended adding more practical, examples from physical or engineering.

Remove the duplicates from the references, [1] and [4] are duplicates.

Standardizing the way of references are written must choose one format.

Finally this paper introduced a distinguished scientific contribution in the field of fractional differential equations.

Reviewers' comments:

Reviewer's Responses to Questions

**Comments to the Author**

1. Is the manuscript technically sound, and do the data support the conclusions?

Reviewer #1: Yes

Reviewer #2: Yes

2. Has the statistical analysis been performed appropriately and rigorously?

Reviewer #1: I Don't Know

Reviewer #2: Yes

3. Have the authors made all data underlying the findings in their manuscript fully available?

Reviewer #1: Yes

Reviewer #2: Yes

4. Is the manuscript presented in an intelligible fashion and written in standard English?

Reviewer #1: Yes

Reviewer #2: Yes

Reviewer #1: I have reviewed your paper " A Comparative Framework for Convergence Analysis of Perturbation Series Techniques in Nonlinear Fractional Quadratic Differential Equations" and I have some notes and suggestions as the following:

1-The manuscript presented in an intelligent form and written in clear and standard English and the methodology is clear

2-The abstract is clear, but it could be shortened.

3-In page 3 some symbols have been written in two way, such as p is inconsistently written as ρ and p in this paper.

4-Recommended adding more practical, examples from physical or engineering.

5-Remove the duplicates from the references, [1] and [4] are duplicates.

6-Standardizing the way of references are written must choose one format.

Finally this paper introduced a distinguished scientific contribution in the field of fractional differential equations.

,

Therefore I recommend acceptance after minor revisions.

Reviewer #2: The abstract is clear, but it could be shortened.In page 3 some symbols have been written in two way, such as p is inconsistently written as \rho and p in this paper.Recommended adding more practical, examples from physical or engineering.

**Do you want your identity to be public for this peer review?** For information about this choice, including consent withdrawal, please see our Privacy Policy

Reviewer #1: No

Reviewer #2: No

---

## [Author Response · Author response to Decision Letter 1]

13 Nov 2025

I have carefully addressed all the reviewers’ comments and completed the revisions accordingly. I sincerely thank the reviewers and the editorial team for their valuable feedback, which has significantly improved the manuscript.

---

## [Editor Report · Decision Letter 1]

17 Nov 2025

A Comparative Framework for Convergence Analysis of Perturbation Series Techniques in Nonlinear Fractional Quadratic Differential Equations

PONE-D-25-50040R1

Dear Dr. Hashim,

We’re pleased to inform you that your manuscript has been judged scientifically suitable for publication and will be formally accepted for publication once it meets all outstanding technical requirements.

Kind regards,

Mohammed Jasim Mohammed Alfahdawi

Academic Editor

PLOS ONE
---

## [Editor Report · Acceptance letter]

PONE-D-25-50040R1

PLOS One

Dear Dr. Hashim,

I'm pleased to inform you that your manuscript has been deemed suitable for publication in PLOS One. Congratulations! Your manuscript is now being handed over to our production team.

Kind regards,

on behalf of

Dr. Mohammed Jasim Mohammed Alfahdawi

Academic Editor

PLOS One